# The Role of AMPK Signaling in Brown Adipose Tissue Activation

**DOI:** 10.3390/cells10051122

**Published:** 2021-05-06

**Authors:** Jamie I. van der Vaart, Mariëtte R. Boon, Riekelt H. Houtkooper

**Affiliations:** 1Laboratory Genetic Metabolic Diseases, Amsterdam Gastroenterology, Endocrinology, and Metabolism, Amsterdam Cardiovascular Sciences, Amsterdam UMC, University of Amsterdam, Meibergdreef 9, 1105 AZ Amsterdam, The Netherlands; jamievandervaart@gmail.com; 2Department of Medicine, Division of Endocrinology, Leiden University Medical Center, 2333 ZA Leiden, The Netherlands; 3Leiden University Medical Center, Einthoven Laboratory for Experimental Vascular Medicine, 2333 ZA Leiden, The Netherlands

**Keywords:** AMPK signaling, brown adipose tissue, energy metabolism, obesity

## Abstract

Obesity is becoming a pandemic, and its prevalence is still increasing. Considering that obesity increases the risk of developing cardiometabolic diseases, research efforts are focusing on new ways to combat obesity. Brown adipose tissue (BAT) has emerged as a possible target to achieve this for its functional role in energy expenditure by means of increasing thermogenesis. An important metabolic sensor and regulator of whole-body energy balance is AMP-activated protein kinase (AMPK), and its role in energy metabolism is evident. This review highlights the mechanisms of BAT activation and investigates how AMPK can be used as a target for BAT activation. We review compounds and other factors that are able to activate AMPK and further discuss the therapeutic use of AMPK in BAT activation. Extensive research shows that AMPK can be activated by a number of different kinases, such as LKB1, CaMKK, but also small molecules, hormones, and metabolic stresses. AMPK is able to activate BAT by inducing adipogenesis, maintaining mitochondrial homeostasis and inducing browning in white adipose tissue. We conclude that, despite encouraging results, many uncertainties should be clarified before AMPK can be posed as a target for anti-obesity treatment via BAT activation.

## 1. Introduction

Obesity is turning into a pandemic and is a growing public health concern worldwide. It has been estimated that around 1.3 billion people were diagnosed with obesity in 2016, and these numbers are still increasing [1]. Despite the lengthy debate about whether obesity can be classified as a disease, a medical consensus regarding this condition has been reached, categorizing obesity as a chronic progressive metabolic disorder by the American Medical Association (AMA) [2,3,4,5]. Besides being a metabolic disease itself, obesity also increases the risk of other associated metabolic disorders, such as type 2 diabetes mellitus (T2DM), hypertension, and cardiovascular diseases [4,5]. Treating obesity has proven to be a difficult task due to the multifactorial nature of the disease. Environmental factors (diet, temperature, physical and psychological stress, sleep quality, and physical activity), genetics, and epigenetics all play a role in the onset of obesity, though the main cause is a chronic imbalance between the amount of energy absorbed from food and the energy expended [5,6]. 

One of the central metabolic organs that play a role in the maintenance of energy homeostasis (the balance between energy intake and energy expenditure) is the adipose tissue, which can roughly be categorized into white adipose tissue (WAT) and brown adipose tissue (BAT) [7]. WAT is primarily noted for being an energy storage depot, producing cytokines and an extensive amount of hormones, releasing energy in the form of free fatty acids (FAs), and playing a role in glucose metabolism [8]. In the case of a positive energy balance, this leads to the accumulation of triglycerides in WAT, eventually causing obesity to emerge. In contrast, BAT has a completely different metabolic function. It is a mitochondria-dense tissue that expresses substantial amounts of uncoupling protein 1 (UCP1) and is therefore believed to be a thermoregulatory organ that is able to generate heat to maintain the core body temperature in case of cold exposure (also called adaptive thermogenesis) [9]. In addition, BAT secretes several factors (e.g., fibroblast growth factor 21, interleukin-6) generally called “BATokines” that may impact metabolism as well [10]. It has been long known that BAT is present in rodents, where it plays a crucial role in nonshivering thermogenesis [11,12]. BAT was considered to have no physiological relevance in adult humans and was assumed essentially nonexistent. We now know this is not true. Using integrated positron emission tomography and computed tomography (PET–CT) in combination with [^18^F]fluorodeoxyglucose, several groups independently demonstrated the presence of cold-activated BAT in adult humans mainly in the neck, clavicular, and spine region [9,11,13]. The energy-combusting capacity of BAT is mainly mediated through the expression of the tissue-specific UCP1 in its mitochondria. The UCP1 protein enables brown adipocytes to dissipate an electrochemical proton gradient through a proton leak in the inner mitochondrial membrane, leading to the uncoupling of oxidative phosphorylation [14]. This uncoupling makes it possible for mitochondria to break down FAs and other energy substrates to generate energy in the form of heat instead of ATP. New discoveries have led to the characterization of an intermediate type of adipocyte, namely beige, or brite, adipocytes in humans. Brite cells show similarities with both white and brown adipocytes. These brite cells reside in WAT and differentiate from certain precursor cells, or by a transition from white to brite adipocytes, a process called “browning.” When unstimulated, these brite fat cells predominantly show WAT-like characteristics, but upon activation, they acquire many mitochondria with high UCP1 expression and high thermogenic capacity [15,16].

Due to its role in energy dissipation rather than energy storage, many efforts have been made to unravel the underlying mechanisms and possible triggers of BAT activation. One major physiological activator of BAT is cold exposure. When a subject is exposed to cold, the hypothalamus will send signals through the sympathetic nervous system (SNS) mediated by norepinephrine that binds the β-adrenergic receptors (β3 receptor in mice and likely β2 receptor in humans) on BAT [17]. This process is subject to several models of physiological regulation, such as the circadian clock [18]. For instance, when the daily rhythm was changed, metabolic processes in skeletal muscle, liver, and BAT were disrupted, leading to an altered glucose and lipid metabolism [19]. More specifically, preclinical studies have shown that prolonged daily light exposure decreases BAT activity through reduced sympathetic outflow to β3-adrenergic receptors on BAT and decreased phosphorylation of AMP-activated protein kinase (AMPK) in BAT [20,21,22,23,24]. Altogether, these events lead to reduced FA and glucose uptake by BAT [24]. 

A second mode of BAT regulation is through intracellular signaling. Among the various signaling pathways that converge on BAT differentiation and activation, AMPK takes a prominent place [25,26]. AMPK is a serine/threonine kinase that has been identified as a sensor for intracellular energetic stress [27,28]. Moreover, AMPK has broader functions involved in whole-body energy balance via the SNS [21]. Although many studies suggest a link between the AMPK pathway and BAT activation, evidence identifying AMPK as a causal factor for BAT activation is still ambiguous. Especially the mechanisms that regulate activation of BAT via AMPK remain unclear. Therefore, in this review, we will focus on the potential of several AMPK-linked pathways as a target for BAT activation. We will dissect the brain–AMPK–BAT axis into individual components and analyze how these can interact to promote thermogenesis. First, we will discuss the molecular mechanism of AMPK activation, as well as the different compounds that are able to activate AMPK. Secondly, we will explore the potential mechanisms by which AMPK can activate BAT. We distinguish three processes that contribute to BAT function, namely, (1) development of BAT, (2) mitochondrial health, and (3) browning. Finally, we will discuss the future of BAT as a potential anti-obesity target.

## 2. AMPK Activation through a Wide Variety of Different Compounds

Previous research has identified the serine/threonine kinase AMPK as an important metabolic switch, with orthologues in different species, that regulates multiple processes involved in lipid metabolism [26,27,29,30,31]. AMPK consists of three subunits: an α-subunit that has catalytic activity, a β-subunit containing a glycogen-sensing domain, and a γ-subunit with a regulatory site for binding to AMP (activating) and ATP (inhibitory). Inactivation of AMPK by ATP will lead to inactivation of catabolic processes that generate ATP, whereas activation of AMPK by AMP will lead to inhibition of anabolic processes that consume ATP and activation of catabolic processes [32]. By virtue of this energy-sensing capacity, AMPK can act as a sensor for disturbances in energy balance caused by metabolic stresses at the cellular and whole-body level. Metabolic stressors can be divided into pathological and physiological stresses. Various pathological stresses, such as glucose deprivation, hypoxia, oxidative stress, and ischemia, cause an increase in the AMP:ATP ratio through reduced ATP production. On the other hand, physiological stresses, including exercise and contraction in skeletal muscles, lead to an increase in the AMP:ATP ratio through ATP consumption [33,34,35]. Upon activation by either of these metabolic stresses, AMP binds the α-subunit of AMPK, after which a conformational change is induced that enhances the activity of AMPK and allows the phosphorylation of a threonine residue (Thr-172) by specialized kinases. Additionally, binding of AMP to the γ-subunit will inhibit dephosphorylation of the Thr-172 thereby prolonging activation of AMPK [22]. This phosphorylation enables AMPK to activate downstream targets, leading to catabolic pathways to be switched on and anabolic pathways to be switched off [36]. As a result, more ATP will be generated or less consumed, contributing to more energy expenditure. Studies over the past two decades have provided critical information on upstream activators of AMPK in vivo, namely, metabolic stress, LKB1, CaMKK2, hormones, and small molecules. We briefly discuss these but also refer the reader to excellent reviews on these topics [28,30,33]. 

The first mode of regulation involves kinases that activate AMPK through phosphorylation. Assays showing AMPK kinase (AMPKK) activity and Western blot analysis of multiple cellular kinases revealed AMPKK activity of liver kinase B1 (LKB1), suggesting LKB1 to be a major kinase in the upstream AMPK cascade [37]. When mouse embryonic fibroblast cells lacking LKB1 were treated with 5-aminoimidazole-4-carboxamide riboside (AICAR), an AMP-mimic, there was a complete lack of Thr-172 phosphorylation, indicating that there was no AMPK activation. However, the mouse embryonic fibroblast cells did show minor activation of acetyl-CoA carboxylase (ACC), a downstream target of AMPK, suggesting that LKB1 is not the only AMPKK [37].

CaM-dependent protein kinase kinase (CaMKK) was identified as a second protein kinase able to activate AMPK. CaMKK itself is activated through increased levels of intracellular Ca^2+^. Although CaMKK seems to have weaker interaction with AMPK than LKB1, several studies have revisited the role of CaMKK over the past decade. The CaMKK has two isoforms—CaMKKα and CaMKKβ. Although the small molecule CaMKK inhibitor STO-609 inhibits AMPK activation in LKB1^-/-^ HeLa cells, it is nonspecific for the isoforms of CaMKK. When CaMKKβ is experimentally silenced using siRNA, this results in the full reduction of the activation of AMPK. In contrast, silencing of CaMKKα led to no significant effect [38,39], supporting the notion that AMPK can be activated by LKB1 and CaMKK, especially the CaMKKβ isoform. 

A second mode of regulation is through compounds. The mechanism by which these compounds activate AMPK can be divided into three classes, namely, (1) increasing levels of AMP, (2) mimicking one of the in vivo activators, and (3) directly activating AMPK by binding one of the subunits [33]. Although it is beyond the scope of this review to give a detailed description of all of these compounds, we will briefly touch upon a selection of interesting compounds (see also, Figure 1 and Table 1). Metformin, a widely prescribed antidiabetic drug, lowers glucose levels in the blood and inhibits Complex I of the respiratory chain, which is a vital enzyme in the electron transport chain. Additionally, metformin can mimic glucose deprivation, which causes transient energy stress in the cell that will contribute to AMPK activation through the first-mentioned mechanism [36,40,41,42]. Moreover, many mitochondrial poisons and plant-derived drugs, such as oligomycin, resveratrol, and berberine, are able to activate AMPK through respiratory chain inhibition [33,43,44]. As for AMP-mimicking drugs, the best described AMP-analogue is AICAR, which can be converted intracellularly into its monophosphate form (called ZMP), consequently mimicking the AMP effects on AMPK [43,45]. Finally, the third class can be characterized by a large group of small molecules that can bind to AMPK directly. One example is A769662, a thienopyridine that can allosterically activate AMPK and prevents dephosphorylation of Thr-172 through binding to the Allosteric Drug and Metabolite (ADaM) site, located between the β-CBM and the N-lobe on the α-subunit [23,46,47,48].

In conclusion, different kinases and many compounds are able to activate AMPK in different manners [33,35,37,38,39,40,43,46,47,49]. These regulators can contribute to metabolic health via different signaling cascades, all coming down to activation of AMPK, which will subsequently trigger catabolic processes, including increased FAO, glycolysis, and glucose uptake, processes that are fundamental for BAT activation. 

## 3. The Functional Mechanism of Regulation of Brown Adipose Tissue by Hypothalamic AMPK 

Cold activates BAT by increasing sympathetic nervous system (SNS) outflow from the hypothalamus toward BAT. Several hypothalamic neurons that are responsible for the regulation of thermogenesis and energy expenditure are located in the arcuate, dorsomedial (DMH), paraventricular, and ventromedial (VMH) nuclei and in the lateral hypothalamus (LHA) and preoptic (POA) area [64,65]. Upon cold exposure, the POA is activated and projects to neurons in the VMH, which will lead to inactivation of AMPK and subsequent sympathetic outflow toward BAT [64,66,67]. The link between AMPK (in)activation and BAT involves two separate mechanisms, namely, (1) via downregulation of AMPK in the hypothalamus (Figure 2A) or (2) through activation of AMPK intracellularly in brown adipocytes (Figure 2B). While AMPK activity in brown adipocytes results from a local energy deficit, hypothalamic AMPK is mainly a reflection of whole-body energy balance and increases energy levels through SNS outflow to metabolic organs, such as BAT [22,26]. Inhibition of hypothalamic AMPK induces a signaling cascade that consequently causes norepinephrine (NE) to be excreted in the vicinity of brown adipocytes. NE binds to β2-adrenergic (human) or β3-adrenergic (rodents) receptors on the surface of brown adipocytes, activating intracellular AMPK, increasing intracellular cAMP, and phosphorylating protein kinase A (PKA), which will induce intracellular lipolysis in BAT. In addition to cold, several compounds and hormones are also suggested to alter BAT thermogenesis through their effect on hypothalamic AMPK activity in the VMH [22]. In this section, we will discuss the function of hypothalamic AMPK and its regulation by different stimuli. 

### Hormones, Compounds, and Metabolites Regulate AMPK in the Hypothalamus

The thyroid axis was identified as an important regulator of lipid metabolism and whole-body energy balance [68]. In a condition called hyperthyroidism, excess thyroid hormones are produced characterized by increased energy expenditure and weight loss. In rats, injection of T3 in the VMH raises energy expenditure through inactivation of hypothalamic AMPK, stimulation of SNS outflow toward BAT, and activation of thermogenesis [68,69]. This effect was reversed by blocking the β3-adrenergic receptors, blunting the expression of thermogenic markers in BAT, and preventing reduction in body weight associated with T3 administration [68]. Steroidogenic factor 1 (SF1) neurons in the VMH are responsible for the projection toward the SNS and subsequently BAT in response to T3 treatment since mice lacking SF1 neurons have abnormal VMH development and are obese [26,70,71]. It is thought that SF1 neurons in the VMH play a role in the regulation of energy balance by integrating multiple signals both peripherally and centrally [71,72,73,74]. Deletion of AMPKα1 in SF1 neurons leads to a similar effect as observed after T3 injection, including weight loss and increased UCP1 levels in BAT [71]. However, when mice lacking UCP1 were treated with T3, hypothalamic AMPK was not affected, suggesting UCP1 is needed for the T3-mediated effect of thermogenesis [75]. Furthermore, central administration of T3 in rats is associated with increased browning markers of WAT, which was also mediated by hypothalamic AMPK in the VMH [26,75,76]. Similar to thyroid hormones, estrogens also play an essential role in energy expenditure. Estradiol levels decrease during menopause in women, leading to decreased energy expenditure and weight gain [77]. When estradiol was administered in hyperphagic rats lacking estradiol, hypothalamic AMPK was inactivated, thermogenic markers in BAT were upregulated and energy expenditure was increased [78]. When the rats were subsequently treated with the AMPK activator AICAR, the anorexic effect evoked by estradiol was diminished. Similar to results found with T3, blockage of β3-adrenergic reversed the effects of estradiol, increasing body weight and reduced UCP1 expression in BAT [78]. These results suggest that estradiol is indeed able to regulate hypothalamic AMPK, thereby activating BAT thermogenesis. Of note, both thyroid hormone and estradiol are thought to be able to reduce stress in the endoplasmic reticulum, leading to less ceramide synthesis in the VMH, subsequently inhibiting AMPK and promoting BAT activation [71,79]. 

Nicotine, an important constituent of cigarette smoke, also influences energy expenditure, mediated by inhibition of hypothalamic AMPK in the VMH and subsequent sympathetic outflow to BAT [80,81]. Research has also pointed out that peripheral nicotine treatment in rats led to increased expression of UCP1 in BAT [82,83]. This may at least partly explain why nicotine withdrawal after cessation of smoking is associated with reduced energy expenditure.

Multiple studies have identified that several bone morphogenetic proteins (BMPs), part of the growth factor β superfamily, are key actors in thermogenesis. BMPs have pleotropic effects in different tissues. In adipose tissue, BMP2, BMP4, BMP7, and BMP8B are associated with the differentiation of preadipocytes to brown adipocytes [84,85,86,87,88]. BMP8B is expressed in BAT and the hypothalamus and is associated with increased thermogenesis [66,89]. Mice lacking BMP8B have impaired thermogenesis, while BMP8B treatment led to an elevated response to adrenergic stimulation with norepinephrine [89]. Stereotaxic administration of BMP8B specifically in the VMH in rats also decreased hypothalamic AMPK, increased thermogenic markers in BAT, and increased browning markers in WAT [90]. This effect was not found when BMP8B was administered in the LHA, suggesting the effects are specific for the VMH. To support further the hypothesis that BMP8B only has an effect on AMPK in the VMH and not the LHA, AMPKα was overexpressed in the VMH with a constitutive promotor, which completely abolished the BMP8B effect on thermogenesis [89]. In contrast, when a dominant-negative form of AMPKα was expressed in the VMH, there was a greater thermogenic effect after central administration of BMP8B. Combined, these data suggest that BMP8B is able to increase thermogenesis in an AMPK-mediated manner in the VMH. 

Glucagon-like peptide-1 receptor (GLP-1R) agonists are widely used in clinical practice as therapy for type 2 diabetes and obesity and one of its beneficial effects is that it induces a negative energy balance by decreasing appetite. However, we recently showed that treatment of healthy lean men with the GLP-1R agonist exenatide enhanced [^18^F]FDG uptake by BAT as measured by PET–CT scan, pointing to increased BAT activity [91]. In rats, injection of GLP-1 in the DMH increases core body temperature and expression of thermogenic markers [92]. Central administration of the GLP-1R agonist exendin-4 increases sympathetic outflow toward BAT, leading to increased UCP1 protein levels, reduced lipid content, and increased FA uptake [93]. However, it is not clear if the effects of GLP-1 and exendin-4 on BAT thermogenesis are mediated by AMPK signaling. Nevertheless, the GLP-1R agonist liraglutide is able to inhibit hypothalamic AMPK in the VMH [94,95]. In rats, central administration of liraglutide induced weight loss, significantly increased the temperature of BAT, gene expression of thermogenic markers in BAT, browning markers in WAT, and protein levels of UCP1 in both WAT and BAT [95]. In line with the effects observed after BMP8B administration, constitutive expression of AMPKα or AICAR treatment in the VMH of rats attenuated the liraglutide-induced UCP1 expression in BAT, suggesting that the effect of liraglutide on weight loss and UCP1 expression is indeed mediated by AMPK in the VMH [22,95]. 

Lastly, it was found that metabolites also influence hypothalamic AMPK [67]. For example, α-lipoic acid (α-LA), a short-chain FA that is a cofactor of several enzymes, reduced body weight in rats by increasing energy expenditure and gene expression of *Ucp1* in BAT and WAT [96]. Furthermore, administration of α-LA inhibited hypothalamic AMPK and this effect was diminished after activation of AMPK with AICAR or after overexpression of AMPKα in the brain. In obese subjects, 8-week administration of α-LA resulted in a moderate reduction of body weight [97,98]. Nevertheless, these results did not elucidate the mechanism behind the weight loss; thus, it is not clear if this is due to increased energy expenditure, and whether AMPK was involved. 

In summary, there is a large group of different compounds, ranging from hormones and growth factors to small molecules and metabolites that regulate hypothalamic AMPK ref. [22,26,33,44,57,68,69,99]. As a general concept, these compounds downregulate hypothalamic AMPK in the VMH and activate BAT-mediated thermogenesis. In the next section, we will discuss the role of AMPK in the activation of brown adipocytes in more detail. 

## 4. The Functional Mechanism of AMPK-Mediated Activation in Brown Adipocytes

In contrast to the role of AMPK in the hypothalamus preventing a negative energy balance by reducing the sympathetic outflow, AMPK plays a very different role in BAT. In BAT, AMPK is directly activated by (1) cold exposure that stimulates β-adrenergic signaling, (2) kinases, (3) compounds, as described in Section 1 and Figure 2, and (4) by downregulation of hypothalamic AMPK [100]. Intracellularly, in brown adipocytes, AMPK can phosphorylate adipose triglyceride lipase (ATGL) and hormone-sensitive lipase (HSL), enzymes that promote intracellular lipolysis, again via PKA. This generates free FAs that can bind to UCP1, thereby causing a conformation change that activates the protein, which leads to dissipation of the electrochemical gradient across the mitochondrial inner membrane and will promote heat production in mitochondria, though studies have shown that AMPK may only have a minimal role in adipocyte lipolysis [100]. Additionally, AMPK can promote translocation of the cluster of differentiation 36 (CD36) and activation of lipoprotein lipase (LPL) that play an important role in extracellular lipolysis and subsequent uptake of triglyceride (TG)-derived fatty acids from circulating lipoproteins. Lastly, AMPK can phosphorylate ACC, thereby releasing the inhibition of carnitine palmitoyltransferase 1 (CPT1) at the mitochondria (Figure 1). The CPT1 protein is responsible for FA transport into mitochondria, providing fuel for oxidative phosphorylation and dissipation of heat by UCP1. This role is particularly clear in the liver. However, in mice with an ACC knockin mutation, AMPK phosphorylation and removal of CPT1 inhibition did not influence BAT thermogenesis [100]. It is hypothesized that AMPK is required for the acute activation of BAT thermogenesis, but the effect of AMPK on ACC may have a more inferior role in BAT thermogenesis as previously thought.

Studies with adipocyte-selective deletion of the α-subunit or the β-subunit have shown that AMPK is essential in multiple processes in BAT, including mitochondrial structure and function, energy expenditure, and brown adipocyte development [23,100,101]. In this section, we will discuss activation and regulation of BAT via the AMPK axis by means of three important processes: (1) development of BAT, (2) mitochondrial health, and (3) browning (Figure 3).

### 4.1. Development of Brown Adipocytes

The existing body of research on the relation between BAT and AMPK suggests an important role for AMPK in the development of brown adipose tissue. To further elucidate the pathways involved in brown adipogenesis, chemical inhibitors, and siRNA technologies were used to block different AMPK-linked signaling cascades, such as MAPKs, the LKB1–AMPK pathway, the mTOR pathway, and Wnt signaling cascades and effects on brown adipogenesis were studied [102]. Two important upstream and downstream proteins of the AMPK pathways are LKB1 and tuberous sclerosis complex 2 (TSC2), respectively. LKB1 activates AMPK when nutrients are low, which in turn causes phosphorylation of TSC2, giving the cell a stress signal that will lead to inhibition of mTOR, independent of protein kinase B (PKB), also known as Akt [103]. In contrast, mTOR is activated in an Akt-dependent mechanism that negatively regulates TSC2, leading to cell growth and proliferation [104]. Furthermore, AMPK directly regulates mTOR via the phosphorylation of the regulatory-associated protein of mTOR (RAPTOR), which is a binding partner of mTOR, leading to inhibition of mTOR [103,105]. Altogether, these findings suggest a link between AMPK and mTOR, and the next question is whether the AMPK-mTOR cascade is also involved in the development of brown adipocytes. Research suggests that the mTOR pathway is vital for the early stages of brown adipocyte differentiation from preadipocytes to brown adipocytes. In contrast, in the later stages of BAT development, activation of AMPK leads to phosphorylation of TSC2 and RAPTOR, and consequent inhibition of mTOR-mediated proliferation of brown adipocytes [33,102]. Therefore, the AMPK–mTOR axis plays an important role in the time-resolved regulation of brown adipogenesis [33,106]. Multiple studies have shown that mTOR is able to enhance glucose uptake by increased translocation of the GLUT1 transporter in BAT [107,108]. This suggests that sustained inhibition of mTOR through AMPK is not a viable strategy for BAT activation because inhibition might have a negative effect on glucose uptake. 

Epigenetic markers are of great importance in the development of cells from progenitor cells. Methylation and acetylation of key developmental genes are critical processes in the development of all cell types. For brown adipogenesis, a key transcriptional regulator is the PR domain containing 16 protein (PRDM16) [109]. This regulator is able to bind and activate the peroxisome proliferator–activator receptor (PPARγ), an essential DNA-binding transcriptional factor, thereby stimulating adipogenesis and increasing lipolysis. Loss of PRDM16 causes differentiation of brown adipocytes toward skeletal myoblasts, whereas expression of PRDM16 induces differentiation into BAT, additionally suggesting a different cell lineage for BAT and WAT [44]. Further investigation into the regulation of adipogenesis by PRDM16 revealed a key role for AMPK in activation of PRDM16 through α-ketoglutarate (αKG) [110]. Since isocitrate dehydrogenase 2 (IDH2) is responsible for the conversion of isocitrate to αKG in mitochondria and αKG levels are regulated by AMPK, it was investigated whether AMPK regulates IDH2, thereby aiding adipogenesis [111]. It was found that depletion of AMPKα1 reduced intracellular IDH2 activity, leading to less αKG, which is needed for demethylation of the *Prdm16* promotor [110,111]. Furthermore, deletion of the same AMPKα1 subunit resulted in weakened expression of *Prdm16* in BAT progenitor cells, which led to reduced adipogenesis and fewer multilocular lipid droplets [101]. These results suggest that AMPK is needed for PRDM16 expression by reducing IDH2 expression and indirectly decreasing αKG levels and subsequent demethylation of the *Prdm16* promotor.

### 4.2. Mitochondrial Health 

A major part of the thermogenic capacity of BAT is dependent on its mitochondrial abundance. A growing body of research shows an essential role for AMPK in the maintenance of mitochondrial homeostasis [112]. More specifically, in addition to serving a role as an energy sensor, AMPK can also activate transcriptional regulators of mitochondrial biogenesis, mitochondrial fission, and mitophagy [15,110]. It is beyond the scope of this review to give a detailed description of the signaling pathways that underlie the intrinsic regulation of all these processes, but we refer to comprehensive reviews on the relation between AMPK and mitochondrial health for more detail [112,113].

AMPK-activating compounds such as metformin, resveratrol, berberine, and AICAR all have an effect on mitochondrial health in brown adipocytes [33,43,45]. For example, metformin activates AMPK, which in turn inactivates ACC, thereby releasing the inhibition on CPT1 and increasing lipolysis and FAO [22,40,114]. Moreover, metformin is able to phosphorylate HSL at PKA phosphorylation sites, also contributing to lipid breakdown. Resveratrol can activate AMPK, which will subsequently lead to activation of sirtuin-1 (SIRT1) through deacetylation, activation of the peroxisome proliferator-activated receptor-γ co-activator 1 (PGC1α) and to increased mtDNA content, mitochondrial size and abundance, eventually leading to reduced lipogenesis and enhanced FAO [15]. Mice that were treated with resveratrol showed increased expression of both UCP1, SIRT1, and PGC1α in BAT and increased BAT mitochondrial function, all contributing to higher energy expenditure [50,51]. Furthermore, resveratrol prevented high-fat-diet-induced obesity and insulin resistance in mice [51]. Additionally, AMPK-mediated activation of SIRT1 will also lead to activation of PGC1α and PPARγ [15], linking mitochondrial biogenesis to developmental processes in brown adipocytes and lipolysis. Likewise, berberine is able to increase UCP1 and PGC1α expression through an AMPK-mediated pathway. Despite not being fully elucidated, multiple studies suggest that berberine activates AMPK by increasing the AMP:ATP ratio and by inhibiting the mitochondrial complex I of the respiratory chain [44,99]. In parallel to these effects on mitochondria, it was also shown that PRDM16 was indispensable for berberine-induced brown adipogenesis [44]. This suggests that berberine has the potential to activate AMPK-mediated thermogenesis in BAT via two mechanisms, namely, by regulating the expression of key proteins in mitochondrial biogenesis (UCP1 and PGC1α) and by regulating an important epigenetic factor of brown adipogenesis (PRDM16) [44,99]. Although not thoroughly investigated in BAT, AICAR-mediated activation of AMPK has been shown to lead to increased glucose uptake, FA uptake, and insulin sensitivity in skeletal muscle in mice [115]. Since skeletal muscles and BAT are connected through common progenitors and because muscles are able to cause metabolic stress signals through contraction thereby activating AMPK, one could speculate that AICAR might have similar effects on BAT. These findings emphasize that most genes involved in mitochondrial biogenesis are under the regulation of PGC1α and PPARγ [112]. When taking a genetic approach, mice lacking the β-subunit of AMPK showed disrupted cristae and reduced mitochondrial respiration, suggesting defective mitochondrial structure and function [100]. 

The process of mitochondrial fission is vital for the maintenance of a healthy mitochondrial population. Indeed, mitochondrial fission can increase the mitochondrial surface, thereby enhancing the ability of free FAs to bind to and activate UCP1 [116]. Once activated, AMPK is able to induce fragmentation of mitochondria across cell types with morphometrically distinct mitochondrial networks [117]. Mitochondrial fission is mainly regulated through mitochondrial fission factor (MFF), which is a receptor of dynamin-like protein 1 (DRP1) on the mitochondrial outer membrane, an essential protein for mitochondrial fission. Proteomics screens revealed two phosphorylation sites on MFF that are substrates for AMPK. Additionally, activation of AMPK increases the localization of DRP1 in mitochondria [117], thus linking AMPK and mitochondrial fission via phosphorylation of MFF and subsequent activation of DRP1 [108,109]. Lastly, to maintain a healthy network of mitochondria, biogenesis is critical, but it is also essential for old mitochondria to be broken down. Selective autophagy of mitochondria, or mitophagy, is regulated by AMPK-mediated phosphorylation of Unc-51 such as autophagy activating kinase (ULK1) [112]. Inhibition of mTOR by AMPK also relieves the inhibition of ULK1 by mTOR, revealing the great intricacy by which these processes are regulated. However, when looking at the same mouse model mentioned earlier, lacking the β-subunit, the protein levels of PGC1α and phosphorylation levels of MFF were not different from the control mice, suggesting no alteration in mitochondrial number [100]. In contrast, the ULK1 protein show reduced phosphorylation in the AMPKβ knockout mice, compared to the control, leading to less mitochondrial clearance. 

Taken together, AMPK is of vital importance in maintaining BAT by regulation of mitochondrial health. Its contribution in cold-induced thermogenesis in BAT involves regulation of mitochondrial structure, function, and mitophagy but is not essential for mitochondrial fission. 

### 4.3. Browning 

Brite cells and their role in thermogenesis are only now beginning to be unraveled. Although very similar in function, BAT and brite cells develop in a distinct manner with different signaling cascades [102,106,118]. AMPK is a key regulator in the formation of brite cells located in WAT (browning) [119], as shown through a high-throughput screen combined with small hairpin RNA (shRNA) knockdown [118]. This confirms earlier results when AMPK was silenced by siRNAs in brown adipocytes, showing that brown adipogenesis was completely arrested [102]. AMPK inhibition led to reduced proliferation of brown preadipocytes to mature brown adipocytes [106]. Specific knockdown of the α1 subunit of AMPK led to altered proliferation, favoring fibrogenesis instead of brown adipogenesis, supporting a similar lineage between skeletal muscles and BAT [23,101,106,118]. Moreover, independent knockdown of the AMPK subunits AMPKα1, AMPKα2 and combined knockdown of the subunits AMPKβ1, AMPKβ2, AMPKγ1, and AMPKγ3, led to decreased abundance of UCP1 in white adipose tissue, suggesting reduced browning [110,118]. When mice were treated with the β-adrenergic agonist CL316,243, the mRNA expression of important markers for browning, such as *Cidea, Ppara, Pdk4*, and *Ucp1*, were all upregulated in the control mice but not in the AMPKβ1/AMPKβ2 knockout mice [100]. These data suggest that AMPK is essential for β-adrenergic-mediated browning. AMPK activating compounds also induce browning of WAT toward brite cells [46,47,102]. For instance, when mice were treated with AICAR, they showed increased UCP1 protein abundance in WAT that correlated with the number of brown-like adipocytes [102]. Together, these data suggest an important role for AMPK in WAT browning resulting in the physiological regulation of thermogenesis, although the elucidation of the full mechanism requires additional investigation.

Concluding, AMPK can regulate adipogenesis through a variety of partly intersecting processes including inhibition of the mTOR pathway, demethylation of PRDM16, and activation of PPARγ. AMPK has an important role in mitochondrial homeostasis through signaling cascades involving PGC1α, PPARγ, MFF, and ULK1. Finally, AMPK activation is a major inducer of the browning process of brite cells. Altogether, there is compelling evidence that AMPK is involved in a multitude of developmental and functional processes of BAT and that it can activate BAT via many different signaling pathways. 

## 5. Discussion

The discovery of functional BAT has led to a growing interest in BAT as a therapeutic target for obesity because activation of BAT contributes to energy metabolism, which may result in creating a negative energy balance. The purpose of the current review was to determine whether AMPK could be used as a target for BAT activation. Although there is compelling evidence that AMPK can activate BAT, many facets of the precise mechanism are still unknown, and studies were mainly performed in mouse models. For instance, rather than the classical BAT cells, there may be a prominent role for brite adipocytes upon AMPK activation [110,118]. In addition, BAT volume diminishes with age and obesity, providing another reason to turn to alternative tissues, such as brite cells, to increase energy expenditure. In line with this, the question of whether BAT is a plausible target for the treatment of obesity in humans has been a frequently investigated matter. Unfortunately, AMPK-activating compounds that were mentioned in this review were mostly investigated in cultured cells, mice, or rats, with the exception of resveratrol, berberine, and metformin. This makes it particularly challenging to apply the results as an integrated treatment for obesity, also due to the complexity of this disease, which is hard to mimic with animal models or cell cultures. Nevertheless, there are a few clinical trials published in which BAT activators were tested for their ability to activate BAT in humans. Though most of these studies are short term and therefore cannot draw definite conclusions on their possible therapeutic use as anti-obesity treatment. They do, however, give more insight into the induction of BAT via AMPK and the potential positive effect on humans with obesity. A recent early phase 1 clinical trial tested the effect of mirabegron, a β3-adrenergic receptor agonist, on older human subjects with obesity and insulin resistance, showing that mirabegron substantially improved glucose metabolism and induced browning of WAT (NTC02919176) [120]. Similar results were found in the phase 1 clinical trial with healthy young women (NTC03049462) [121]. Although these findings were not directly linked to activation of BAT via AMPK, the results underline the significance of BAT as a therapeutic target for obesity. Interestingly, besides AMPK, other compounds are also able to increase energy expenditure and lipid homeostasis [122]. Recent research into the hormone FGF21 has indicated that it is able to activate AMPK in adipocytes [122,123]. However, this does not explain the beneficial effects since FGF21 treatment is not dependent on AMPK or ACC, suggesting that FGF21 and AMPK work synergistically [124]. Therefore, combinatorial treatment with FGF21 and AMPK activators may have enhanced therapeutic efficacy as an anti-obesity treatment because of these divergent signaling routes. Finally, additional research investigating brite cells identified UCP1-independent thermogenic pathways regulated by creatine and Ca^2+^ cycling. These compounds can stimulate nonshivering thermogenesis in brite cells, independently of UCP1, through stimulation of the tricarboxylic acid cycle, electron transport chain, and FA reesterification pathways [125,126,127]. There is abundant room for further progress in determining the importance of AMPK in these and other pathways. 

Another factor that complicates the study of in vivo effects of AMPK activation in BAT or browning is the fact that the most used method to activate BAT, namely, cold, also affects other tissues such as skeletal muscle and the liver, and more studies should focus on this intertissue relationship. For example, a study quantifying BAT activation and whole-body glucose metabolism in patients with T2DM after 10 days of cold acclimation showed only a small increase in [^18^F]FDG uptake by BAT despite a marked improvement in glucose metabolism [128]. In a follow-up study, the experiment was repeated, but shivering was prevented, leading to a complete reduction of the favorable effects [129]. This suggests that either muscle contraction (shivering) or a strong cold stimulus is needed to stimulate the positive effects of BAT. The question remains whether the physiological changes can be ascribed to activation of BAT, which is estimated to expend between 25 and 211 kcal/d [130]. It is possible that BAT releases endocrine factors that regulate skeletal muscles [131,132], which would mean that the effect on the muscle is indirect and still requires BAT activation. However, it should be noted that most of the human BAT studies were performed using ^18^FDG–PET–CT scans to measure glucose uptake as a proxy of BAT activation. This is far from an ideal visualization modality in subjects with type 2 diabetes that are severely insulin resistant, which very likely results in less ^18^FDG uptake by metabolic tissues including BAT. Therefore, a tracer based on triglyceride-derived fatty acid uptake is likely superior. Altogether, the debate about the relevance of BAT for energy expenditure and obesity treatment is all but resolved. 

In conclusion, AMPK integrates a wide range of inputs and activates various downstream pathways. There is plenty of evidence connecting AMPK and BAT activation through different signaling cascades. To minimize the side effects when therapeutically targeting AMPK, further research should focus on identifying key effectors of thermogenesis. Many questions remain to be answered before the development of an anti-obesity treatment via AMPK-mediated BAT activation in humans.

## Figures and Tables

**Figure 1 cells-10-01122-f001:**
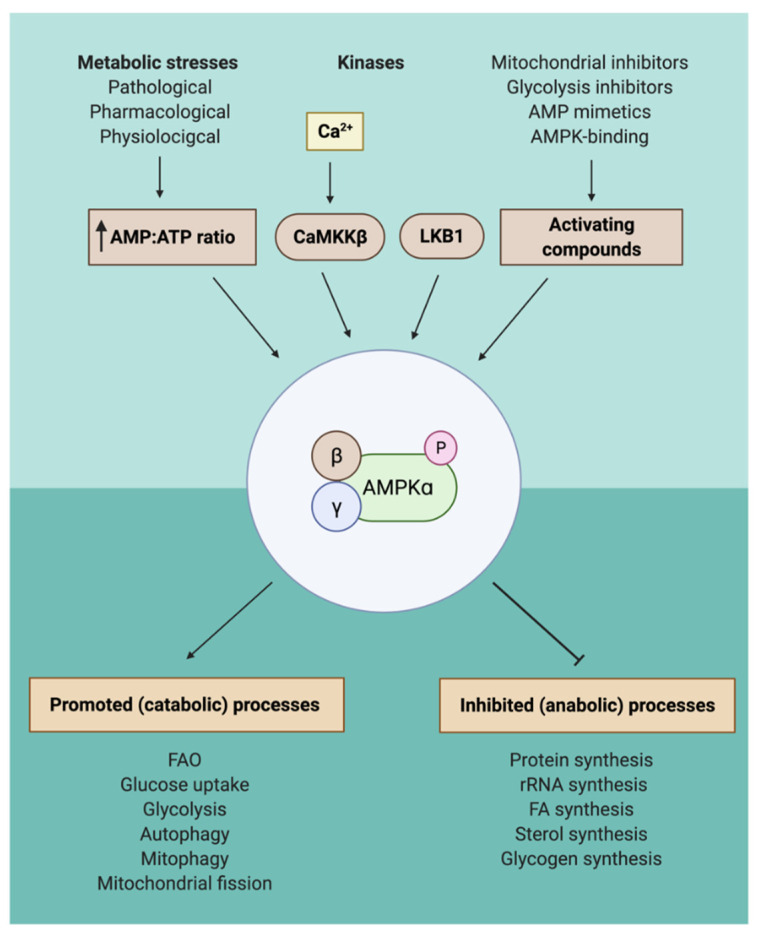
Functional mechanism of upstream regulation of AMPK. Phosphorylation of AMPK can be caused by an increased AMP:ATP ratio, the kinases CaMKKβ and LKB1, and activating components. Elevated ratios of AMP:ATP are caused by metabolic stresses, which can be divided into pathological, pharmacological, and physiological pathological stresses. CaMKKβ is activated by increased intracellular Ca^2+^. Examples of activating compounds are mitochondrial and glycolysis inhibitors, AMP-mimetics, and AMPK-binding compounds. Activation of AMPK will stimulate catabolic processes, leading to ATP production, and inhibit anabolic processes, leading to less ATP consumption. Some examples of the metabolic consequences are shown. Abbreviations: adenosine monophosphate, AMP; adenosine triphosphate, ATP; AMP-activated protein kinase, AMPK; CaM-dependent protein kinase kinase beta, CaMKKβ; fatty acids, FA; fatty acid oxidation, FAO; liver kinase B1, LKB1.

**Figure 2 cells-10-01122-f002:**
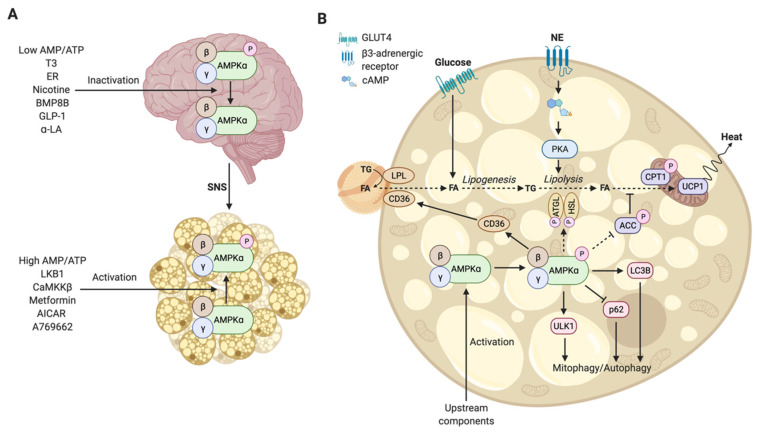
Schematic overview showing the functional mechanism of BAT activation mediated by AMPK. (**A**) Brain–AMPK–BAT axis, where BAT can be activated by inhibition of hypothalamic AMPK, after which the SNS will signal to BAT to upregulate lipolysis, or through direct activation mediated by different upstream components of AMPK. (**B**) Detailed overview of BAT activation through norepinephrine, excreted by the SNS, or through direct activation of AMPK. Norepinephrine can bind the β3-adrenergic receptor, which will subsequently activate cAMP and PKA, ultimately increasing lipolysis and heat production. Upstream components of AMPK, previously described, activate AMPK after which CD36 is translocated to the membrane and LPL is stimulated, resulting in increased uptake of triglyceride-derived FA from lipoproteins and later in lipolysis combined with heat production. In addition, AMPK can release the inhibition of CPT1, increasing FA transport to mitochondria, also contributing to heat production, though this latter part of the signaling cascade is likely to only play a minor role in BAT thermogenesis. The role of AMPK in lipolysis and phosphorylation of ACC has not been proven experimentally; therefore, dashed lines are used in this figure. Abbreviations: acetyl–CoA carboxylase, ACC; 5-Aminoimidazole-4-carboxamide ribonucleotide, AICAR; alpha-lipoic acid, α-LA; adenosine monophosphate, AMP; AMP-activated protein kinase, AMPK; adipose triglyceride lipase, ATGL; adenosine triphosphate, ATP; bone morphogenetic protein 8B, BMP8B; cyclic AMP, cAMP; CaM-dependent protein kinase kinase beta, CaMKKβ; cluster of differentiation 36, CD36; carnitine palmitoyltransferase 1, CPT1; estrogen receptor, ER; fatty acid, FA; glucagon-like protein 1, GLP1; hormone-sensitive lipase, HSL; low-density lipoprotein, LDL; liver kinase B1, LKB1; protein kinase A, PKA; sympathetic nervous system, SNS; triglycerides, TG; triiodothyronine, T3; uncoupling protein 1, UCP1.

**Figure 3 cells-10-01122-f003:**
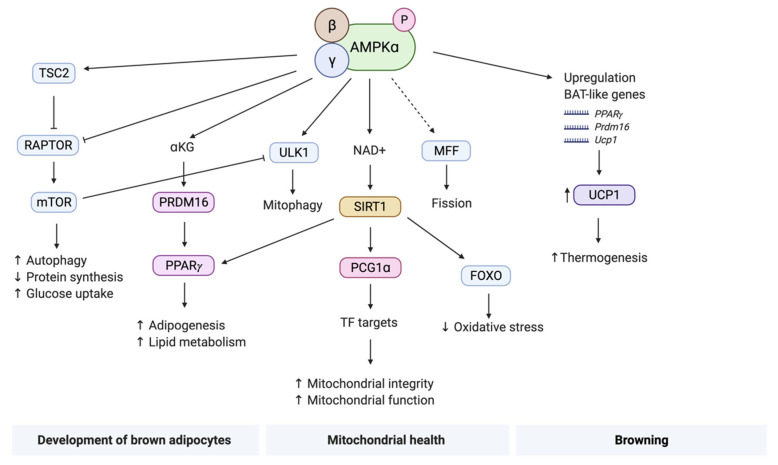
Selection of AMPK-mediated signaling cascades involved BAT activation. The pathways are divided into three processes, namely, the development of brown adipocytes, mitochondrial health, and browning. The signaling cascades depicted here are simplified and are far more complex and interconnected than shown. Firstly, mTOR is a major kinase involved in autophagy and protein syntheses in the developmental stage and glucose uptake in adult brown adipocytes. Secondly, PRDM16 is an essential DNA-binding transcription factor, which can indirectly induce adipogenesis and lipolysis. Thirdly, SIRT1 is an important kinase involved in mitochondrial health and also connects development and mitochondrial homeostasis by influencing the transcription factor PPARγ. Finally, an abundance of the UCP1 protein is associated with increased thermogenic capacity and is indirectly regulated by AMPK. The role of AMPK in the phosphorylation of MFF has not been proven experimentally; therefore, a dashed line is used in this figure. Abbreviations: AMP-activated protein kinase, AMPK; α-ketoglutaric acid, αKG; forkhead box protein O, FOXO; mitochondrial fission factor, MFF; nicotinamide-adenine-dinucleotide, NAD; peroxisome proliferator-activated receptor-γ co-activator 1, PGC1α; peroxisome proliferator–activator receptor, PPARγ; PR domain containing 16 protein, PRDM16; regulatory-associated protein of mTOR, RAPTOR; sirtuin-1, SIRT1; transcription factor, TF; tuberous sclerosis complex 2, TSC2; uncoupling protein 1, UCP1; Unc-51 like autophagy activating kinase, ULK1.

**Table 1 cells-10-01122-t001:** Compounds that activate AMPK. Compounds can be divided into two classes of indirect activators and direct activators, depending on their mechanism of action and binding site [48]. Secondly, AMPK activators can be subclassified into mitochondrial inhibitors, glycolytic inhibitors, AMP mimics, and AMPK-binding compounds.

Compound Name	Mechanism of Action	Binding Site	Type	Function	Reference
**Metformin**	Indirect activator	AMP binds γ subunit	Mitochondrial inhibitor	Mimics glucose deprivation by lowering the glucose blood levels. Inhibits Complex I of the respiratory chain causing transient energy stress in the cell, elevating the AMP:ATP and ADP:ATP ratio.	[26,40,41,42,49]
**Oligomycin**	Indirect activator	AMP binds γ subunit	Mitochondrial inhibitor	Mitochondrial poison that activates AMPK through inhibition of Complex V of the respiratory chain, elevating the AMP:ATP and ADP:ATP ratio.	[33,43]
**Resveratrol**	Indirect activator	AMP binds γ subunit	Mitochondrial inhibitor	Plant-derived drug that activates AMPK through inhibition of Complex I, III, and V of the respiratory chain, elevating the AMP:ATP and ADP:ATP ratio.	[33,43,50,51,52,53]
**Berberine**	Indirect activator	AMP binds γ subunit	Mitochondrial inhibitor	Plant-derived drug that activates AMPK through inhibition of Complex I of the respiratory chain, elevating the AMP:ATP and ADP:ATP ratio.	[33,43,44,54]
**2-deoxy-glucose**	Indirect activator	AMP binds γ subunit	Glycolysis inhibitor	A non-metabolizable glucose analog. Elevates the AMP:ATP and ADP:ATP ratio by suppressing glycolysis.	[33,49,55,56]
**AICAR**	Direct activator	ZMP binds γ subunit	AMP mimetic	Pro-drug that is converted to ZMP, mimics energy stress.	[22,33,57]
**A769662**	Direct activator	Binds the ADaM site. Selective for AMPKβ1 isoform.	AMPK binding	Small molecule that activates allosterically by binding AMPK. Prevents dephosphorylation of Thr-172.	[23,46,47,48]
**PF-249**	Direct activator	Binds the ADaM site. Selective for AMPKβ1 isoform.	AMPK binding	Small molecule that activates allosterically by binding AMPK. Prevents dephosphorylation of Thr-172.	[48,58,59]
**PF-739**	Direct activator	Binds the ADaM site. Act on all 12 mammalian AMPK isoforms.	AMPK binding	Small molecule that activates allosterically by binding AMPK. Prevents dephosphorylation of Thr-172.	[58]
**MK-8722**	Direct activator	Binds the ADaM site. Act on all 12 mammalian AMPK isoforms.	AMPK binding	Small molecule that activates allosterically by binding AMPK. Prevents dephosphorylation of Thr-172.	[60,61]
**PF-06409577**	Direct activator	Binds the ADaM site. Selective for AMPKβ1 isoform.	AMPK binding	Small molecule that activates allosterically by binding AMPK. Prevents dephosphorylation of Thr-172.	[48,59,62,63]

Allosteric drug and metabolite, ADaM; adenosine diphosphate, ADP; adenosine monophosphate, AMP; AMP-activated protein kinase, AMPK; adenosine triphosphate, ATP.

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
