# Peer review of "The Role of AMPK Signaling in Brown Adipose Tissue Activation"

_cells, 2021, doi:10.3390/cells10051122_

Round 1

Reviewer 1 Report

The review here presented by van der Vaart and co-workers described the current, increasing, knowledge about the potential role of AMPK-mediated signaling in BAT activation and how this could be hypothetically used as a target for anti-obesity.

The review is well written and quite comprehensive, and I applaud the authors in-depth coverage of the bibliography. Nevertheless, the review is less comprehensive on some parts of the review (i.e the current understanding of the role of AMPK in browning, a process which currently drives a high interest in the scientific community).

Some major a minor points should be addressed previous final acceptance for publication in Cells.

Major comments:

1) Although the authors indicates that “it is beyond the scope of this review…” I strongly encourage the authors to include a table reporting the specific inhibitors and compounds mentioned in Figure 1, together with the phenotype induced and the reference. This would be very helpful to the reader.

2) In Figure 2, the scheme represents the phenotype of WAT cells/tissue (big lipid droplet, with the nucleus located at the cell periphery) rather than BAT. The informatic tool used by the authors to perform the scheme have available better schemes to represent BAT than those used in the current version of the manuscript.

3) When describing “Browing” in section 3, the authors indicate: “AMPK is a key regulator in the formation of brown adipocytes from WAT….”. This affirmation is not enterally true and can induce misunderstanding. WAT and BAT cells derived from different progenitors. Beige or brite cells located in WAT can differentiate to BAT but not WAT itself. This process is detailed described by Perdikari et al., 2018, BATLAS: Deconvoluting Brown Adipose Tissue. Please clarify this point.

Minor comments:

  • In the first phrase of the Abstract as well as in the Introduction, the expression “global epidemic” could be replaced by “pandemic”.
  • In section 2, line 122, Liver Kinase B1 is wrongly abbreviated as LBK1. UCP1 is also referred as UPC1 (section 3, line 297)
  • In Page 6, line 287, the authors describe “… suggests that berberine has the potential to activate AMPK-mediated thermogenesis in BAT via two mechanisms, namely by regulating the expression of key proteins in mitochondrial biogenesis and by regulating an important epigenetic factor of brown adipogenesis”. Could the authors name which the key proteins are??.
  • Meaning of some abbreviations such as ZMP should be indicated.
  • References are sometimes inserted, but not merged, into the manuscript (line 241, 304).
  • In Figures 2 and 3, abbreviations such as ER, GLP-1, a-LA, TF…should be defined in the corresponding figure legends.
  • In Discussion section, when authors describe the current clinical trials using BAT activators, the phase as well as the identificatory of the clinical trial should be indicated.

Reviewer 2 Report

Understanding the functional role and identifying the players in BAT-regulated thermogenesis is a real challenge to find news to combat obesity. This is an area of vigorous research where AMPK has emerged as a promising target. In this review, van der Vaart and colleagues provide an overview of recent knowledge on AMPK function in BAT. This is divided into 3 main sections describing the role of AMPK during development of BAT, in maintaining mitochondrial health in BAT and for the browning of BAT. Although the authors have well documented the involvement of AMPK within BAT intracellular signaling, they should include a more detailed part on the major role of hypothalamic AMPK in the modulation of BAT thermogenesis through its regulatory action on the SNS. It is important to highlight that activation of AMPK in BAT by stimuli that increase energy demand is in contrast to AMPK’s role in the hypothalamus, where it prevents negative energy balance by reducing sympathetic outflow (for review see Eur J Clin Invest. 2018 Sep; 48(9): e12996). In addition, there are also important caveats and some issues have not been correctly addressed regarding the role of AMPK in BAT. The authors should present a more balanced view of the current literature. The piece could be developed in a more critical review of recent publications on the role of AMPK in BAT function (e.g., lipolysis, FAO, mitochondrial biogenesis/ function). There are obvious omissions of relevant topics and recent papers are not discussed at all.

Major points:

1. The authors do not mention Mottillo et al, 2016, Cell Metabolism 24:118. In this important paper, mice with knock-out of AMPK beta1/2 subunits were studied. The results suggest that
- AMPK has a minimal role in regulating adipocyte lipolysis. Therefore, it seems perverse to show regulation of BAT lipolysis through ATGL and HSL phosphorylation in Figure 2B.
- The role of AMPK in adipose tissue thermogenesis does not appear to involve acute regulation of fatty acid oxidation. By using mice with an ACC1-S79A and ACC2-S212A knockin mutation (ACC DKI), it was shown that AMPK-mediated phosphorylation of ACC and subsequent removal of CPT1 inhibition was not involved in BAT thermogenesis. Thus, these results indicate that AMPK is required for acute BAT thermogenesis, but this effect does not depend on AMPK phosphorylation of ACC. Folowing on previous comment, Figure 2B should be revised.
- AMPKb1b2AKO mice have altered mitochondrial morphology and function, along with a defect in autophagy signaling. However, BAT mitochondrial number was not altered in AMPKb1b2AKO mice nor was there an alteration in MFF phosphorylation. Thus, it seems that the contribution of AMPK in cold-induced thermogenesis is associated with its role in the maintenance of mitochondrial quality (through the regulation of mitophagy and the clearance of old/damaged mitochondria) rather that mitochondrial biogenesis and dynamics.

2. Cite and discuss data from these recent studies:
- Mol Metab. 2017 Jun; 6(6): 471–481 showing that the effects of FGF21 administration are independent of adipocyte AMPK and were not associated with changes in browning of white (WAT) and brown adipose tissue (BAT).
- Cao et al., 2020, Acta Biochimica et Biophysica Sinica, 53, 112 showing that AMPK controls Idh2 expression via suppressing H2B O-GlcNAcylation.

3. The authors state in the summary of the second section that "there is a large group of different compounds that activates AMPK, ranging from growth factors and transcription factors, to small molecules and hormones", without citing any literature to back this up and there is no mention of the various mechanisms in this section.

4. There are many sections that are too speculative and should be revised according to the current knowledge on AMPK in BAT coming from studies using AMPK KO mouse models (Mottillo et al, 2016, Cell Metabolism 24:118; Wu et al., Front Physiol. 2018, 9: 122; Zhao et al., 2017, Biochem Biophys Res Commun. 2017, 491:508; Cao et al., 2020, Acta Biochimica et Biophysica Sinica, 53, 112).

Minor points:
- mention β-adrenergic stimuli activate AMPK in human brown adipocytes (Mottillo et al, 2016, Cell Metabolism 24:118)
- describe AMPK catalytic and regulatory subunits expression profile in BAT (Mottillo et al, 2016, Cell Metabolism 24:118; Wu et al., Front Physiol. 2018, 9: 122) and discuss the data from adipocyte-selective deletion of AMPKa1 and AMPKb1b2 KO in mice (Mottillo et al, 2016, Cell Metabolism 24:118; Zhao et al., 2017, Biochem Biophys Res Commun. 2017, 491:508).

Overall, I would recommend that these issues need to be discussed in the context of this review.

Round 2

Reviewer 1 Report

In their revised version of the manuscript, the authors have responded to the requiered questions (both major and minor points) then largely improving their manuscript, which can be now accepted for publication.

Just few minimal typos should be corrected:

Line 31: “Obesity is turning into an pandemic..

Line 225: “Hormones, , compounds and metabolites regulate AMPK in the hypothalamus”   

Reviewer 2 Report

This MS is substantially improved and nearly ready for publication. The manuscript has been revised according to my suggestions. However, I have noted inaccurate informations in figures and tables that may confuse some readers, and I would recommend to include these minor changes:
    â—¦    Figure 1: Gluconeogenesis should be removed from the list of the metabolic pathways down-regurated by AMPK activation. Recent reports demonstrated that AMPK is not a direct regulator of hepatic gluconeogenesis (PMID 28467931; 28107516; 20577053).
    â—¦    Figure 3: why gluconeogenesis is mentionned has a feature of BAT function? Similarly, why mitochondrial biogenesis is mentionned as a consequence of AMPK activation in BAT. This is in contrast to the results from Mottillo et al, 2016, Cell Metabolism 24:118. "Mitochondrial biogenesis" should be changed to "Mitochondrial integrity" or "Mitochondrial quality". In the figure legend, describe the meaning of dashed lines.

    â—¦    - Figure 2, Part A & B: why the authors show 2 (P) on activated AMPK (in contrast only one (P) is shown n Figure 1)? the critical regulatory phosphorylation site is Thr172 on the α subunit. 
    â—¦    Figure 2, Part B: in this figure showing the effects mediated by AMPK activation, it remains unclear why the authors still show AMPK-dependent regulation of BAT lipolysis through ATGL and HSL or BAT FAO through phosphorylation of ACC despite this has not been proven experimentally (Mottillo et al, 2016, Cell Metabolism 24:118). To avoid confusion for the readers, these signaling pathways should be simply removed or the figure legend should be revised to clearly mention the meaning of dashed lines. In addition, the critical role of AMPK for maintaining mitochondrial integrity and function in BAT is not clearly shown here.
    â—¦    Figure 1 & 3: AMPK complex is represented as a trimeric complex but only the β and α subunits are indicated. What about γ subunit? In addition, a phosphorylation mark (P) is shown on the third subunit (AMPKγ ?) but this should be located on the α subunit.
    â—¦    - Table 1: in the table legend, it should be indictaed that the different compounds activating AMPK can be classified regarding their mechanism of action. The list of compounds could be divided in at least 2 main sections (see Table 1 in PMID 28546359):
    •    indirect AMPK activators that act by increasing cellular AMP:ATP and ADP:ATP ratios 
    •    Direct AMPK activators acting by binding the γ subunit or the ADaM site located between the β-CBM and the N-lobe on the α subunit.

In addition, the authors should include last generation direct AMPK activators such as MK-8722; PF- 249, PF-739 (PMID 28705990; 28467931) and indicate they can act as pan- or AMPK-complex-selective compounds (e.g., A769662 and PF-249 behave as AMPKß1 selective compounds, in contrast to MK-8722 and PF-739 act as allosteric activator of all 12 mammalian AMPK complexes). 

I regret the authors have not cited the work published in Mol Metab. 2017 Jun; 6(6): 471–481 because it shows that adipocyte AMPK or its downstream substrate ACC are not involved in the beneficial metabolic effect of FGF21 in BAT suggesting that FGF21 does not share a similar mechanism to AMPK activators and that combinatorial treatment with FGF21 and AMPK activators may have enhanced therapeutic efficacy since they function through different signaling pathways. Therefore, these data could be cited in the discussion when tackling the significance of BAT as therapeutic target for obesity.
